# Plasma-Sprayed Flexible Strain Sensor and Its Applications in Boxing Glove

**Yongsheng Liao [1], Yue Cheng [2], Zhongyu Zhuang [2], Rongjun Li [3], Yuan Yu [2], Ruixue Wang [2] and Zhiwei Jiao [2,\*]**

1. PE College, Hunan University of Arts & Science, Changde 415000, China
2. College of Mechanical and Electrical Engineering, Beijing University of Chemical Technology, North Third Ring Road 15, Chaoyang District, Beijing 100029, China
3. Beijing Satellite Manufacturing Co., Ltd., Beijing 100094, China
* Correspondence: jiaozw@mail.buct.edu.cn

**Abstract:** The most common and easy approach to fabricating flexible strain sensors is based on the deposition principle. To improve the design of the sensing layer pattern, the reproducibility of the process and the sensitivity of the sensor, a controllable low-temperature-plasma spraying method for conducting nanoparticles was proposed. A flexible strain sensor was developed with multiwalled carbon nanotubes as the sensing layer and silica gel films as the substrate. The effects of plasma treatment on the cyclic stability and sensitivity of the sensor were examined and compared. The changes in the sensitivity of the sensor with the pattern parameters were also studied. The sensitivity of the sensor treated with low-temperature plasma was greatly improved (from 3.9 to 11.5) compared to that of an untreated sensor. In addition, pattern parameters significantly affected the rate of change in the resistance. A portable smart boxing glove prototype was developed using the prepared sensor and was then tested. The results showed that the smart glove could transmit and monitor a striking force of 49–490 N in real time with a sampling time, resolution, response time, and recovery time of 100 ms, up to 1.05 kg, 8 ms, and 150 ms, respectively.

**Keywords:** flexible sensor; two-dimensional-plane controllable spraying technology; smart boxing gloves; portable; wireless data transmission

## 1. Introduction

Flexible and stretchable electronic devices, which contain flexible sensors, can be used in wearable, lightweight, portable, biocompatible, economical, and environmentally friendly products. Flexible sensors can be attached to the surface of objects and measure the strain distribution on the detection surface. Thus, they are widely used in several applications, such as bionic skin [1–4], health monitoring in sports [5–9], and communications [10–12]. The design and manufacture of the sensing layer is the most important aspect of flexible resistance-strain sensors. Several methods are used for the preparation of these sensing layers, such as deposition, soft lithography, screen printing, and inkjet spray printing.

Kim et al. [13] obtained a stable and stretchable EGaIn strain sensor (Figure 1a) by printing a liquid metal on a platinum-catalyzed silicone elastomer (Ecoflex™ 00-30) to fabricate the sensor. The sensor exhibited a nonlinear sensitivity and a maximum gauge factor (GF) of <8. Jun Shintake et al. [14] used an elastomer composite filled with carbon black as the sensing layer, while the composite sensing layer was mass-produced using a thin-film casting technology and $CO_2$ laser ablation technology, which facilitated the rapid multilayer fabrication of resistive flexible sensors. The fabricated sensor exhibited a high strain amplitude, low hysteresis, and high repeatability (Figure 1b). However, its GF in the resistive mode was 1.62–3.37. As shown in Figure 1c, Khalid H R et al. [15] used a very simple flexible sensor preparation scheme, which involved using a tape as a mask and a hand-held spray can for spraying. However, the GF (1–1.2) of the prepared sensor

was not high. Jin [16] also used a spray method to manufacture CNT coating films as a sensing layer and study the hysteretic dependence on strain under cyclic loading. Chirst et al. used 3D-printing technology to prepare a flexible sensor with a filament made from a TPU/MWCNT mixture and explored how directional sensitivity was affected by the type of pattern design [17]. Kim et al. [18] fabricated an integrated glove sensor using a direct ink-writing method, and this integrated manufacturing method could improve manufacturing efficiency.

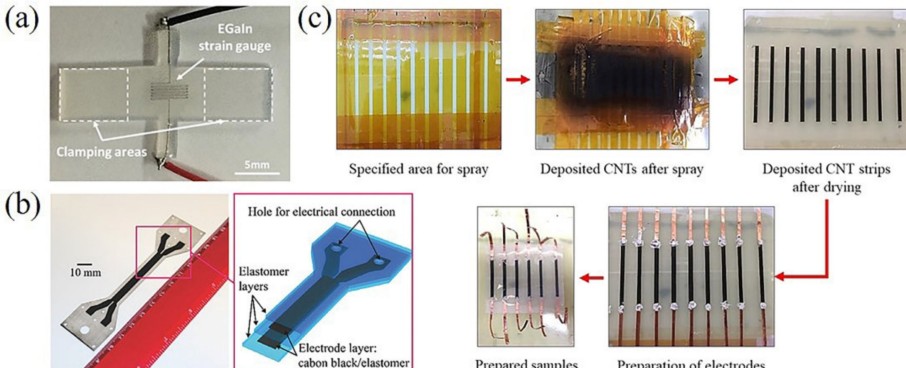

**Figure 1.** (**a**) An EGaIn strain sensor, (**b**) a carbon black/elastomer composite flexible sensor, and (**c**) the fabrication process of a mask-sprayed flexible sensor.

Previous studies indicated that flexible strain sensors prepared using deposition methods exhibited several issues, such as an insufficient bonding force between the sensing layer and the substrate, as well as easy peeling of the sensing layer, which reduced the sensitivity of the fabricated flexible sensors. Therefore, in this study, a method using low-temperature plasma to treat the sprayed material during the spraying process was proposed to improve the sensitivity of the flexible sensors. Here, multiwalled carbon nanotubes (MWCNTs) and silica gel (Ecoflex 0050) were used as the sensing layer and substrate, respectively, of the flexible, stretchable resistive-strain sensors. The effects of low-temperature-plasma treatment on the performance of the flexible sensor, the effects of pattern parameters on the sensitivity of the flexible sensor, and other properties of the flexible sensor were studied.

Boxing gloves are an essential part of the protective gear used in martial arts and fighting competitions. They protect the hands of the boxer and allow them to hit the opponent more efficiently while giving the opponent a certain degree of protection [19]. The development of smart gloves that can monitor the hitting force in real time during hitting will help the judges in determining the hitting force and technique. This will create a fairer environment for boxing competitions and relieve the pressure of enforcing the rules by the judges.

Numerous scholars throughout the world have carried out research work on boxing intelligence. Zhao Jianmin et al. [20] virtualized the real training process using somatosensory interaction and recorded training information in the information system, which provided a new method for virtual boxing training, games, and teaching. Hahn et al. [21] constructed an automatic boxing scoring system by developing a new set of boxing equipment that can be used as a possible solution to the scoring problem for amateur boxers. Noriko et al. [22] designed and developed an innovative smart punching bag as a training tool for boxers. Data on the force applied by the boxer to the punching bag and timing data were recorded, and a smartphone application was developed to transmit the data using a bluetooth to enable coaches and trainees to analyze the training results. At present, the sensors used in the research on boxing sports are basically rigid acceleration sensors, which have a poor human–machine relationship.

A flexible sensor was prepared using a plasma spraying method proposed in this study to be integrated in the boxing glove and combined with the wireless data transmission

technology to achieve real-time monitoring of the hitting force of athletes during boxing matches. In addition, a wireless smart boxing glove was developed and its performance was tested.

## 2. The Controllable Plasma Spraying Method for Conducting Nanoparticles

### 2.1. Low-Temperature-Plasma Spraying Method and Device

To simplify the fabrication method for flexible strain sensors and improve their sensitivity and stability, a programmable controllable low-temperature-plasma spraying method was proposed to fabricate high-sensitivity flexible resistive-strain sensors. Plasma [23] is a fully or partially ionized gaseous state of matter at high temperature or under specific excitation conditions. It is composed of ions, electrons, and a collection of unionized neutral particles. Low-temperature plasma is widely used in the field of material modification due to its low discharge temperature and simple operation. It has a significant effect on local material modification and hardly causes any damage to other structures.

Figure 2 shows the schematic diagram of the low-temperature-plasma spraying device used in this study. It mainly consisted of an aerosol generator, a dual-medium jet device, a three-dimensional (3D) motion platform and its control, a low-temperature-plasma experimental power supply, an argon gas source, a rotameter, connecting wires and pipes.

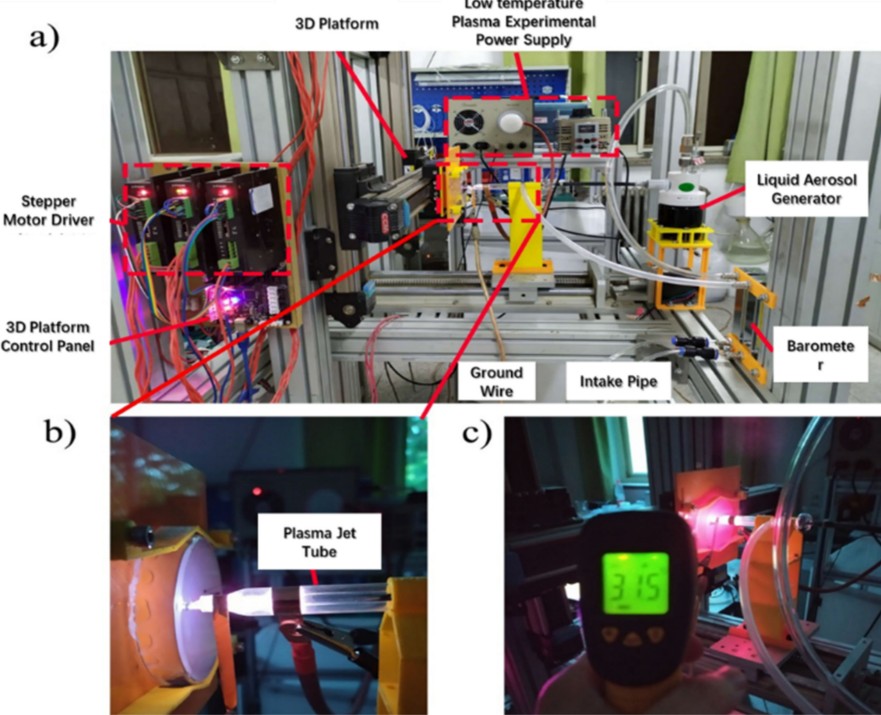

**Figure 2.** The plasma spraying device. (**a**) introduction of the plasma spraying device (**b**) the process of the low-temperature-plasma spraying; (**c**) the temperature of the low-temperature-plasma.

A liquid aerosol generator is a two-fluid atomizer used for generating uniform and stable aerosols with a particle size distribution (i.e., median total aerodynamic diameter, MMAD) of 1–4 μm. Here, a dispersion containing the sensing layer material (CNTs) was placed in the liquid aerosol generator (Figure 2a).

There are three main types of plasma jet generators: single-electrode, double-electrode, and three-electrode structures [24–26]. Because the spray material contains non-working gas components, a dual-medium jet device with coaxial dual jet channels is required. The inner channel sprays the material, and the outer channel generates plasma. Therefore, in this study, a dual-medium jet plasma tube was designed (Figure 2b). The plasma jet took the form of a ring-to-ring discharge. The device consisted of an inner-layer quartz glass

tube, an outer-layer quartz glass tube, a collector, a main air inlet, and an auxiliary air inlet arranged on the jet tube. To enable full contact between the spraying material and the plasma after being ejected from the nozzle, the outer nozzle was extended 2 mm compared to the inner nozzle. Figure 2c indicates that the temperature of the low-temperature-plasma spraying was 31.5 °C.

The control board of the 3D platform was connected to a computer to achieve motion control. The setup was mainly composed of control software, a control board, a driver, a linear slide table, and a DC switching power supply (48 V). The control program was generated by the 3D design and processing software (UG). The Grbl Controller open-source control software was adopted here. The MEGA2560 control board, which is commonly used in 3D printers, was selected for the 3D computer numerical control milling machine control software. Figure 3 shows the 3D platform control interface and the 3D platform control panel. The step speed of the stepping motor we used was 1000 r/min, the speed of the synchronous belt was 0.1~1 mm/s, and the repeatability was 0.05 mm. The movement of the moving part was achieved by using a CCM synchronous belt linear slide table.

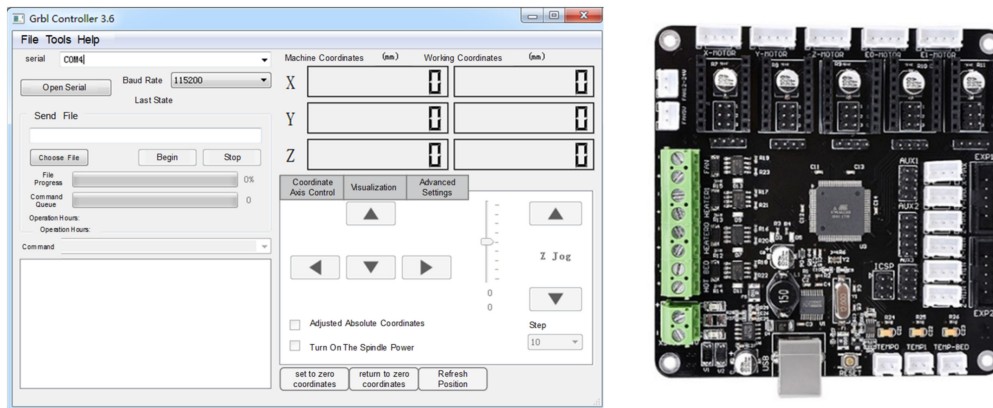

**Figure 3.** The interface of the 3D motion-control software, and the circuit board of the controller.

Figure 4 shows the preparation process for the flexible sensor. The process was as follows. (1) A layer of silica gel was spin-coated on a circular acrylic plate so that the thickness of the silica gel film reached approximately 0.5 mm. (2) The CNT–ethanol dispersion was prepared at a mass fraction of 5%. (3) The path code was generated according to the set path, and the spraying device was then connected to the host computer to start the spraying path program. (4) Next, the spraying process was started, and the 3D platform adhered to the silicone substrate followed the designed path under program control. The air pump was turned on simultaneously, and one channel of argon gas passed through the periphery of the injection tube to generate a plasma arc under the effect of the electric field of the low-temperature-plasma power supply, and the other channel of argon gas entered the liquid aerosol generator to carry the atomized particles from the dual medium. The jet passed through the inner tube of the jet tube and flowed out in the middle of the plasma arc. After contacting the silica gel matrix, the jet firmly adhered to it, forming a predesigned conductive path. (5) After spraying, the electrodes were set at the starting and ending positions of the conductive path, and finally, a 1 mm-thick silica gel film was cured on the surface to complete the encapsulation. The spraying process that does not use plasma to treat the formed sensing layer is called the air-spraying method. Figure 2a shows the device used for this process.

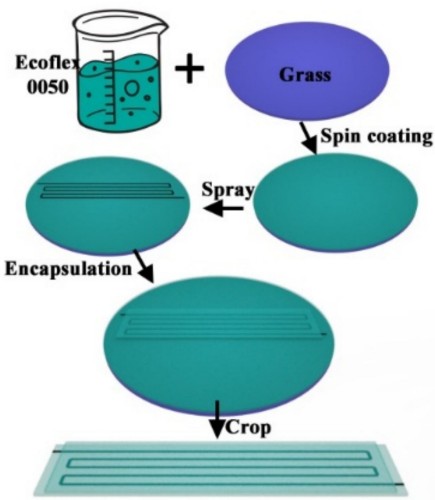

**Figure 4.** The process of fabricating flexible sensors by spraying.

### 2.2. Mechanism of the Low-Temperature-Plasma Treatment

The sensing mechanism of the flexible strain sensor was as follows: when the stretchable base material generated a small strain, the CNT conductive material structure slipped, destroying part of the conductive path and changing the resistance of the sensing layer. In the spraying process proposed here, the use of low-temperature plasma to treat the sprayed material broke some of the carbon–carbon bonds, which reduced the length of the CNTs, and resulted in shorter and more conductive paths. In addition, the adhesion of CNTs to the silica gel substrates was improved. Therefore, under the same strain, the plasma-treated conductive path was easier to break than untreated path, which resulted in more stable changes in the resistance and an improvement in the sensitivity.

## 3. Materials, Manufacturing, and Testing

Figure 5 shows a flow chart for manufacturing smart gloves, after which each part will be introduced separately.

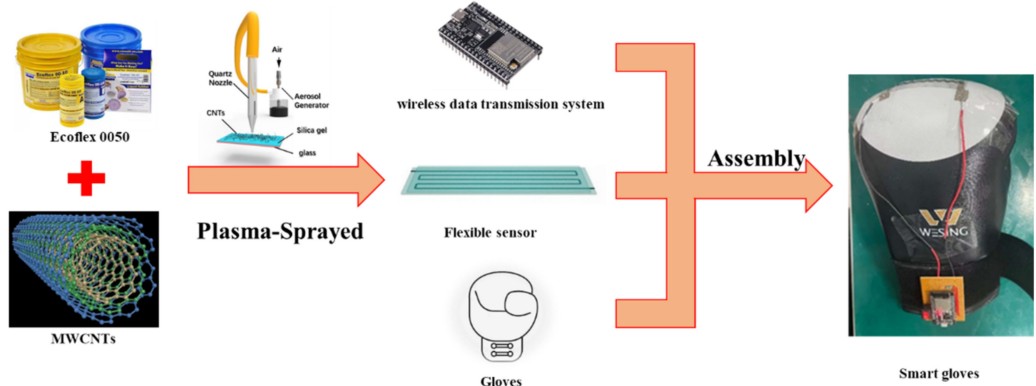

**Figure 5.** Flow chart for fabricating smart gloves.

### 3.1. Experimental Materials and Equipment

#### 3.1.1. Experimental Materials

The MWCNTs (HQNANO-CNTs-010-0, purity > 95 wt%, inner diameter = 3–5 nm, outer diameter = 8–15 nm, length = 3–12 μm, specific surface area > 233 $m^2$/g) were obtained from Suzhou Hengqi Graphene Technology Co., Ltd. (Suzhou, China).

The silicone (Ecoflex 0050, curing time = 3 h, elongation at break = 980%, tensile strength = 315 psi) was acquired from the Smooth-On Company(Marcokey, Palm Beach Gardens, FL, USA).

The CNT alcohol dispersant, which aided in the dispersion of the CNTs in ethanol, isopropanol, n-butanol, terpineol, and other alcohols, was obtained from XFZ21, Nanjing Xianfeng Nanomaterials Technology Co., Ltd. (Nanjing, China).

3.1.2. Experimental Equipment

The equipments used in this experiment are listed in Table 1.

**Table 1.** The main equipment used in the experiment.

| Device | Product Series | Manufacturer |
|---|---|---|
| **Homemade low-temperature-plasma spraying device** | / | / |
| **Homemade tensile strain gage** | / | / |
| **Low-temperature-plasma experimental power supply** | CTP-2000K | Nanjing Suman, Shenzhen Zhongyitong Technology Co., Ltd. |
| **Liquid aerosol generator** | HRH-WAG6 | Beijing Huironghe Technology Co., Ltd. |
| **Glass rotameter** | LZB-6 | Xiangjin Flow Meter Factory |
| **Desktop multimeter** | Tektronix DMM6500 | Guangzhou Xinjingyou Electronic Technology Co., Ltd. |
| **Electronic balance** | WXL-A66002 | Shenzhen Unlimited Weighing Instrument Co., Ltd. |
| **Boxing gloves** | JRS-12345 | Nine Suns Mountain/Nine Days Mountain |
| **Oscilloscope** | DPO 2024B | Ion Technology Co., Ltd. |

*3.2. Manufacturing*

3.2.1. Fabrication of Flexible Sensors

Flexible sensors with different patterns were fabricated using the proposed controllable plasma spraying method for conductive nanoparticles (Figure 6). For comparison, flexible sensors were also fabricated using a spraying method without the plasma treatment. The air pressure during the spraying process was 3 L/min, the amplitude of the plasma power supply voltage was 8.20 V, and the frequency was 89.7 kHz.

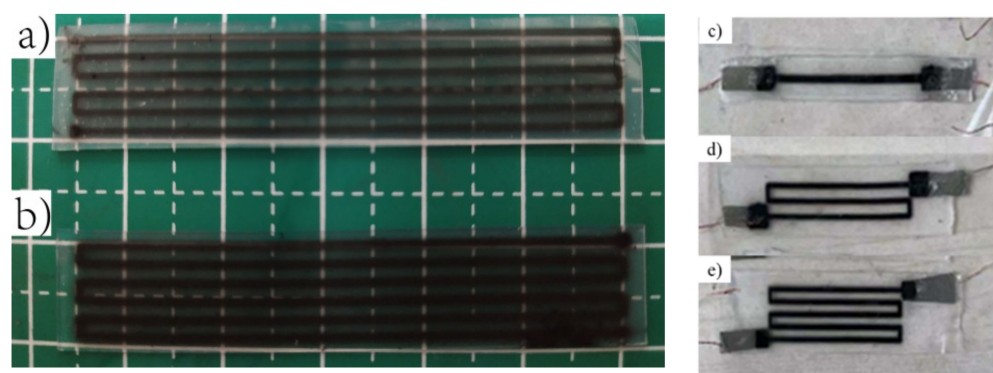

**Figure 6.** Flexible sensors (**a**) sensor without plasma treatment; (**b**) sensor after the plasma treatment; (**c**) sensor with a single sensing path; (**d**) sensor with three sensing paths; and (**e**) sensor with five sensing paths.

3.2.2. Smart Gloves

The prepared flexible strain sensor was closely attached to the surface of the boxing glove. To improve the sensitivity of the sensor and the accuracy of the boxing-force measurement, the path of the sensor was evenly distributed on the fist surface of the boxing glove. Figure 7a shows the optimized glove using multiple folded paths. The data-measurement circuit was mainly composed of an ESP32 development board, a lithium battery, a voltage divider, a flexible sensor, and copper wires. The circuit was fixed on the

boxing glove with velcro tape. Figure 7b shows the prototype of the developed portable smart glove.

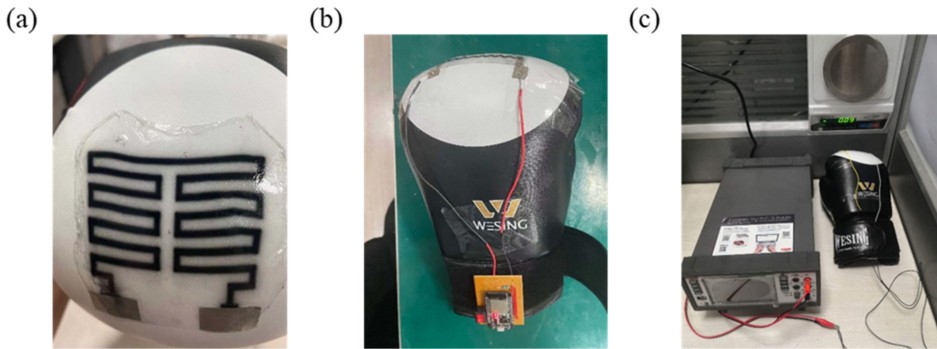

**Figure 7.** (**a**) Flexible sensor path; (**b**) image of the prototype; (**c**) the testing platform.

The Wi-Fi module of the ESP32 development board was set to the soft AP mode as a hotspot to which the host computer was connected. The host computer obtained real-time data of the current by logging in to a webpage with the IP address 192.168.4.1.

### 3.3. Testing

#### 3.3.1. Tensile Test

A tensile-strain gauge that was mainly composed of a stepper motor and a few 3D-printed parts, which were constructed in our laboratory, has been used in the tensile testing and accurately measure the performance of the prepared flexible sensor. Figure 8 shows a schematic diagram of the tensile-strain gauge.

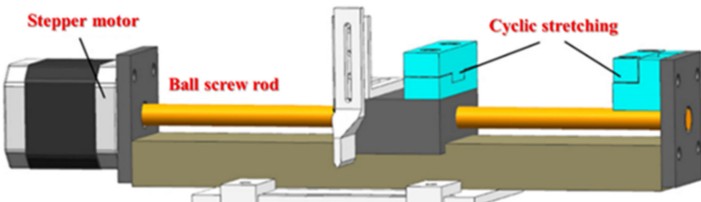

**Figure 8.** Schematic diagram of the developed tensile-strain gauge.

To compare the effect of the plasma treatment, we used a 10% strain rate for stretching. When exploring pattern parameters, we used a strain rate of 10–90% in 10% gradient increments.

#### 3.3.2. Impact Force Test

To accurately simulate the corresponding relationship between the hitting effect of the boxing glove and the sensor resistance of the boxer during competitions, an experimental platform that could measure the relationship between the hitting force and the resistance was built.

The testing platform consisted of a boxing glove equipped with a flexible sensor, a balance, a desktop multimeter, and wires. Figure 9c shows a photo of the testing platform. The impact force was measured by the experimental balance. However, when the glove was struck (Figure 9b), the leather of the glove deformed, which in turn drove the deformation of the conductive nanocoating. This caused a change in the resistance of the conductive nanocoating. Next, the relationship between resistance and time, as well as that between resistance and force, were obtained using a desktop digital multimeter and computer software (Figure 9c).

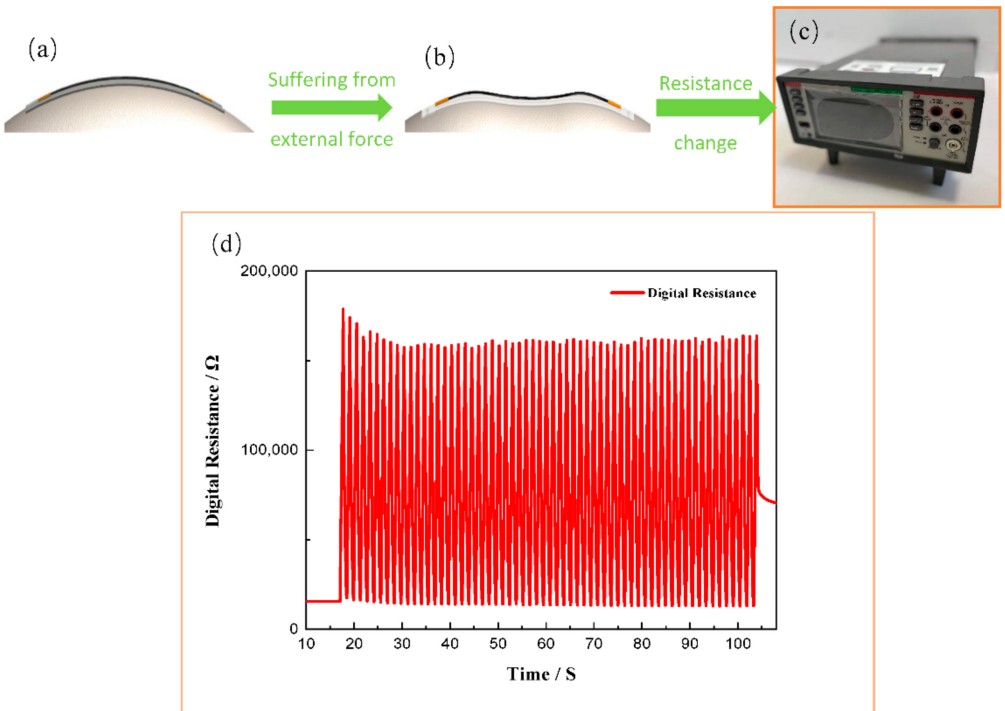

**Figure 9.** The feedback principle of the smart boxing glove: (**a**) before receiving an external force; (**b**) after receiving an external force. (**c**) Desktop multimeter. (**d**) Change in the resistance over time.

## 4. Results and Discussion

### 4.1. Performance of the Flexible Strain Sensor

#### 4.1.1. Sensitivity

The conductive path of the flexible sensor was one or more linear paths formed by conductive nanoparticles. When subjected to an external force, the deformation in the length direction of the conductive path was very clear, and the change in the width direction was small. Therefore, when the original conductive path was subjected to an external force, the path between the conductive particles was reduced (Figure 10). Thus, the change in the external force was represented by the change in the resistance.

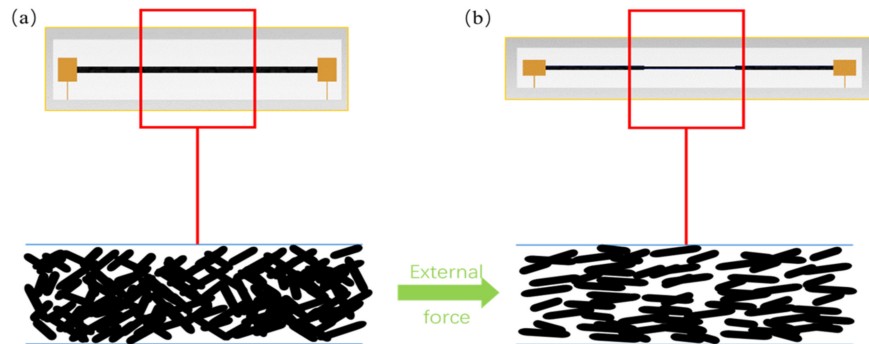

**Figure 10.** Microscopic mechanism of the developed flexible resistive sensor: (**a**) before applying the force; (**b**) after applying the force.

In order to verify the stability and reliability of the strain sensor, a strain extensometer made in our laboratory was used to perform continuous strain–release cycle tests at different strains. The GF of the sensor was calculated based on the resistance change rate and the strain value (Equation (1)).

$$GF = \frac{R - R_0}{R_0 \cdot \delta} \tag{1}$$

where R is the tensile resistance value, $R_0$ is the initial resistance value, and $\delta$ is the tensile strain.

Effect of the Plasma Treatment Process on Sensitivity

Figure 11a,c indicate that the presence or absence of plasma treatment had a significant impact on the electrical properties of the sensing layer. Two resistance-strain sensors (a plasma-treated one and an untreated one) of the same size were stretched several times. The plasma-treated flexible resistance-strain sensor exhibited a relatively stable change in the tensile resistance. The tensile and recovery resistance of the untreated flexible resistive-strain sensor rapidly decreased within the first 300 reciprocating stretches, and the decrease ranged from large to small. After this, the resistance changes stabilized. This indicates that the low-temperature-plasma treatment enhanced the stability of the flexible resistive-strain sensor.

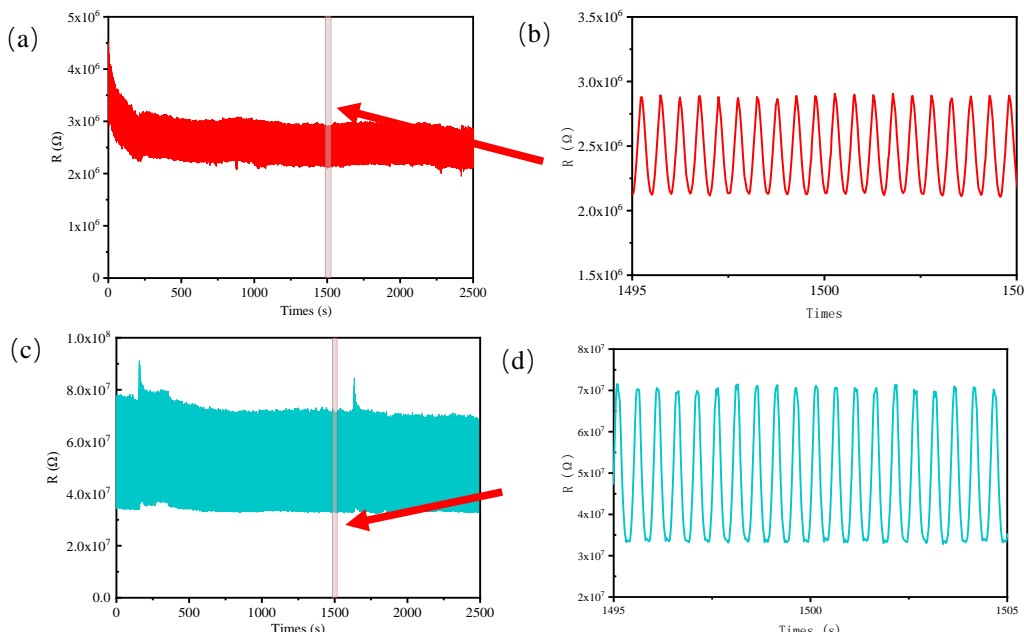

**Figure 11.** Responses of the untreated sensors and the sensors treated with low-temperature plasma to strain: (**a**) the change in the resistance of the untreated flexible sensor stretched 2500 times at 10% strain; (**b**) an enlarged view of the change in the resistance of the untreated flexible sensor; (**c**) the change in the resistance of the flexible sensor after plasma treatment under 10% strain after stretching 2500 times; (**d**) an enlarged view of the change in the resistance of the flexible sensor after plasma treatment.

After the reciprocating tensile resistance values of the two sensors stabilized, they were calculated using the resistance change curve. At a tensile strain of 10%, the GF of the plasma-treated sensor was 11.5, whereas the untreated sensor was approximately 3.9. This indicated that the plasma treatment increased the sensitivity of the sensor because the resistance of the conductive layer changed faster with the strain.

After the low-temperature-plasma treatment, the original structure of the CNTs was destroyed because some carbon–carbon bonds were broken, resulting in a shortening in the length of CNTs. This length-shortening caused some CNTs to lose the ability to form end-to-end conductive paths. Compared with the conductive layers that have not undergone plasma treatment, the conductive paths in the treated conductive layers were reduced, which resulted in macroscopic resistance. During the stretching process of the treated sensor, with the increase in the stretching distance, the number of conductive branches rapidly decreased. This rapidly increased the resistance, which improved the sensitivity of the treated sensor. However, when stretched to a certain extent, no CNTs

could form an effective conductive path with an end-to-end connection, and thus they were no longer conductive.

Effect of the Pattern Parameters on the Sensitivity

Due to the limited length of the paths in the conductive layer that could sense the effective deformation, the decrease in the number of the sensing paths also decreased the effective length of the sensor available for sensing the strain.

Spraying multiple conductive electrode plates in the deformation direction and connecting them end-to-end was equivalent to connecting multiple resistive-strain sensors in series. When detecting a strain at the same place, all sensors detected the strain, and the change in the resistance was amplified. According to the calculation equation of the sensitivity factor, since the same mechanical strain was applied to each sensing path, the change in the resistance rate of each sensing path was also the same under the exact same preparation process. Similarly, their sensitivity factors should have been the same.

Figure 12a shows the variation in the resistance of the sprayed strain sensor with the stretching time. The increase in the conductive path increased the total resistance of the sensor, and the relationship of the resistance change was approximately 1:3:5. The relative change in the resistance of the sensor with five channels was greater than that of the sensor with three channels, which was greater than that of the sensor with one channel. The response to the tensile strain was relatively smooth, with no abrupt increase or decrease and with visible peaks. This indicated that the conductive path prepared by the spraying process had a good response to strain. During the repeated stretching process, the tensile and recovery resistances of all the sensor samples were generally stable, and the shape of the change in the resistance curve remained basically unchanged.

The reciprocating stretching caused the sensor strain to periodically change with time. Figure 12a shows the relative change rate $((R - R_0)/R_0)$ in the resistance versus the time response. The data curves of the three sensors in the figure basically overlap, indicating that although the number of conductive paths laid in the tensile direction was different in the three sensors, their response to strain and sensing characteristic were the same and did not depend on the length of the conductive paths. Thus, these different sensors had the same sensitivity factors. There was no sudden increase or decrease in the curve of the resistance change rate, indicating that the sprayed conductive path in each sensor was continuous and uniform, which further verified the stable performance of the conductive path created by the spraying process.

In addition to the number of paths, we also explored the influence of the three factors of spraying time, length, and width of the same path on the sensitivity. The untreated plasma was investigated at strain rates of 30%, 60%, and 90%, respectively. Figure 12 a shows that the sensitivity first increased and then decreased with the increase in the number of sprays. Figure 12e exhibits the main performance criteria of recently reported typical strain sensors for comparison, indicating the superior performance of strain sensors that were prepared through the plasma-treated controllable-spray method [1,4,9,16,17,27].

4.1.2. Cyclic Stability

Figure 13a,b show the response characteristics of the samples during cyclic stretching under 10% strain and 70% strain, respectively. Under 10% strain, the number of cycles reached 2500 (Figure 11c), whereas under 70% strain, it reached 2000 cycles (Figure 13d). These results showed that the flexible sensor had excellent cycle stability.

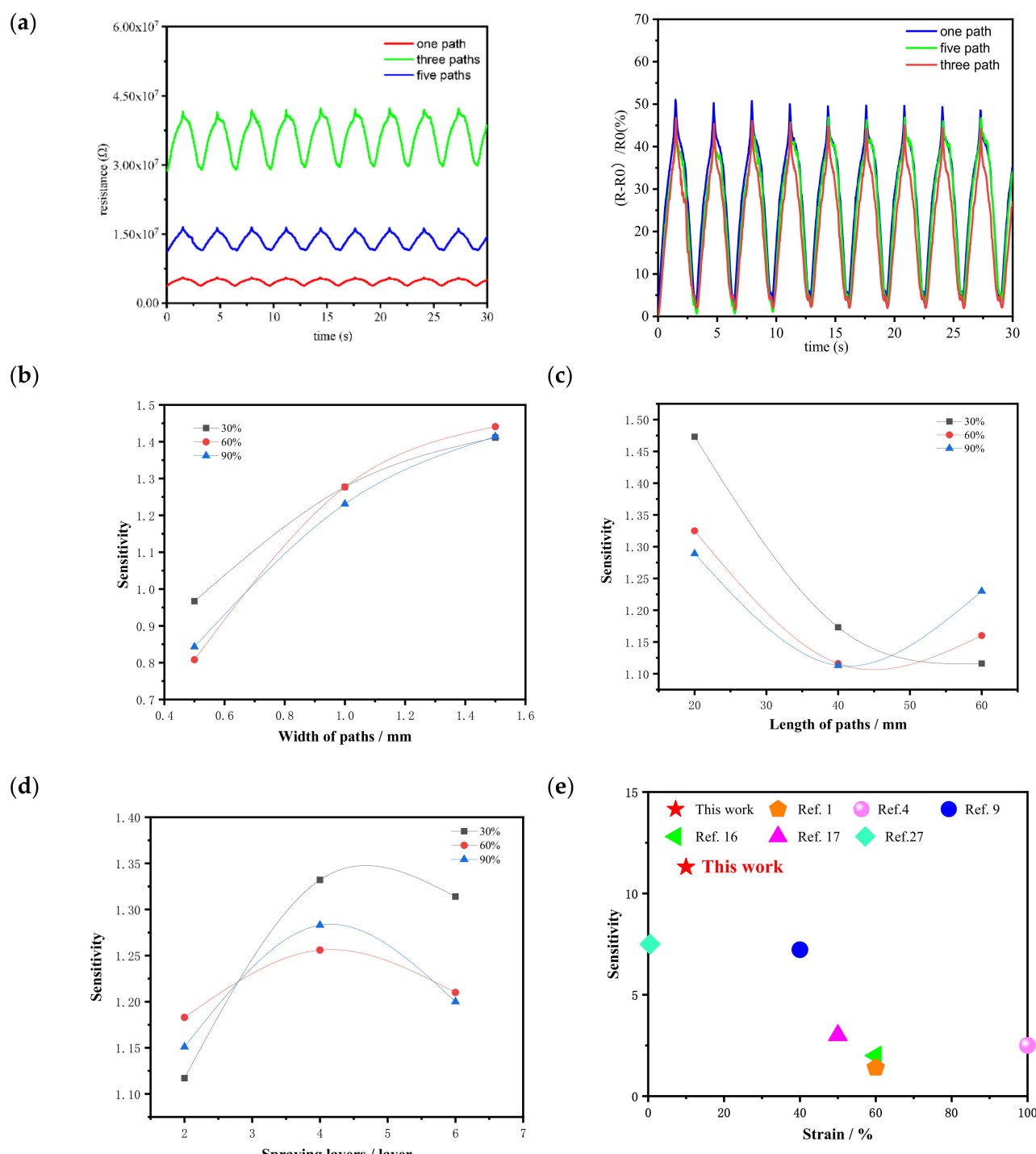

**Figure 12.** Effect of the pattern parameters on the sensitivity of the flexible sensors: (**a**) resistance and resistance change rate of different number paths; (**b**) widths of paths; (**c**) lengths of paths; (**d**) spraying layers; (**e**) comparison of the GF with sensors fabricated using other methods.

### 4.1.3. Strain Range

The flexible sensor after the low-temperature-plasma treatment was stretched, and it showed no conductivity when the tensile strain exceeded 2 mm. This meant that the maximum strain value was 10%. The flexible sensor prepared using a conductive layer without plasma treatment still had good electrical conductivity when the tensile strain reached 70%, indicating that plasma treatment reduced the strain-measurement range of the flexible sensor. After testing, the two sensors reached a near-stationary state at which

the plasma-treated sensor recovered its resistance value, which was 15 times that of the non-plasma-treated sensor. Both flexible sensors showed good resistance changes after stretching. The untreated flexible sensor could retain good resistance even when the tensile strain reached 70% (Figure 13b). However, the treated sensor showing no conductivity and no resistance when it was stretched over 10% strain. This indicated that the treated sensor was suitable for the measurement of a small strain. The relationship between the sensitivity and strain (Figure 13c) indicated that the sensitivity of the untreated sensor decreased with the increase in strain, showing a negative correlation.

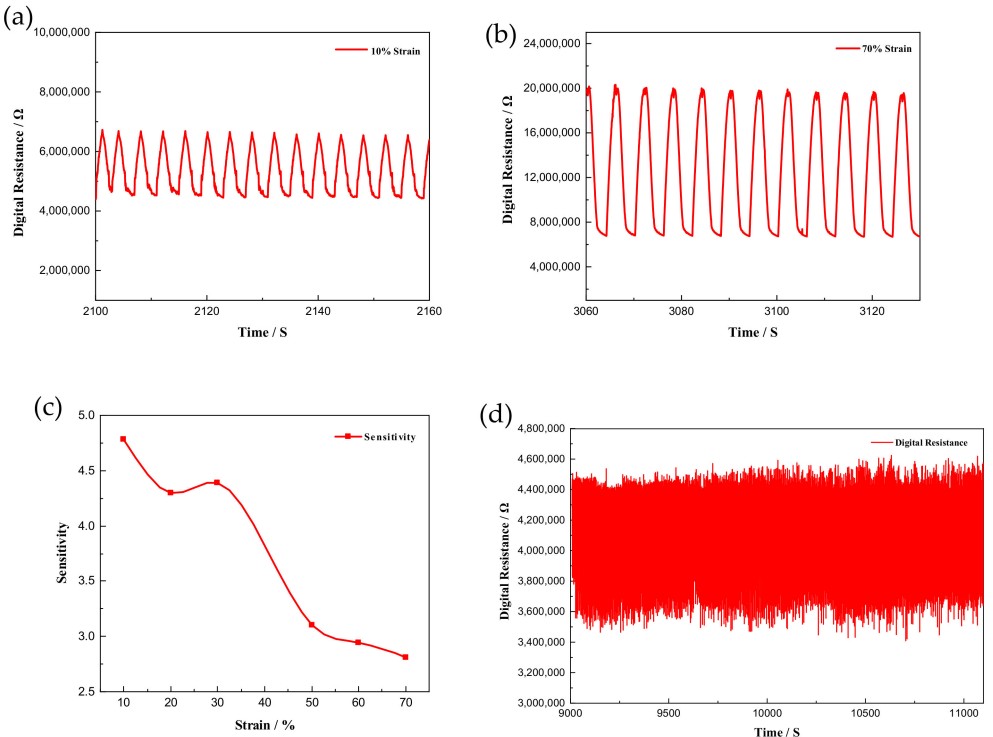

**Figure 13.** (**a**) Changes in the cyclic tensile resistance with time under 10% strain; (**b**) changes in the cyclic tensile resistance with time under 70% strain; (**c**) changes in the sensitivity with strain; (**d**) 70% strain-extension cycle test results.

### 4.2. Performance Test of the Glove Prototype

#### 4.2.1. Wi-Fi Data-Transmission Circuit

The ESP32 development board was set to AP mode. The maximum transmission speed of the board was 150 M/S; however, in the AP mode, the maximum transmission speed was 2 MB/s. The analog-to-digital conversion converted the actual voltage into a digital quantity within 0–4096. Each datum occupied 2 B, so theoretically, 1,000,000 digital data could be read in one second. However, actual testing indicated that one datum per second could form a very smooth curve between the striking force and time, which also improved the data preservation. To test the performance and portability of the portable smart glove, a professional boxing coach was asked to test the prototype in the boxing gym of Hunan University of Arts and Sciences. The results of the on-site test are shown in Figure 14.

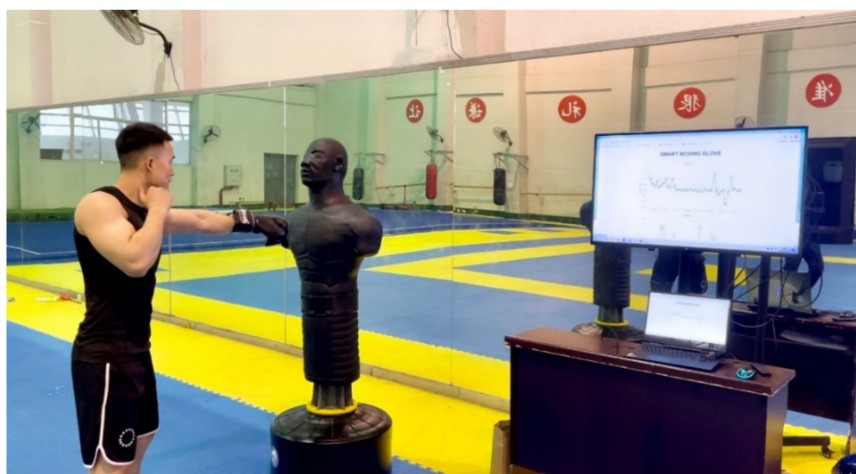

**Figure 14.** Image of the field test.

For easy portability, two 3.7 V lithium batteries were used in series to supply power to the development board. After testing for approximately 1 h, the power supply voltage dropped below 5 V. At this time, the measurement and data transmission circuit stopped working. After 20 min of recharge, the lithium battery continued to work. Thus, the smart boxing glove developed in this article had a battery life of 1 h and a charging time of 20 min.

### 4.2.2. Response and Recovery Times

The performance of the boxing glove was evaluated using the data measured by the testing platform. The data obtained by the multimeter was imported into the computer, and the Origin software was then used to process the data to obtain the relationship between the sensor resistance and time (Figure 15a). To improve the reliability of the data, the measured effective data (Figure 15b) were averaged, and the average response and recovery times of the flexible sensor were calculated to be 100 and 150 ms, respectively.

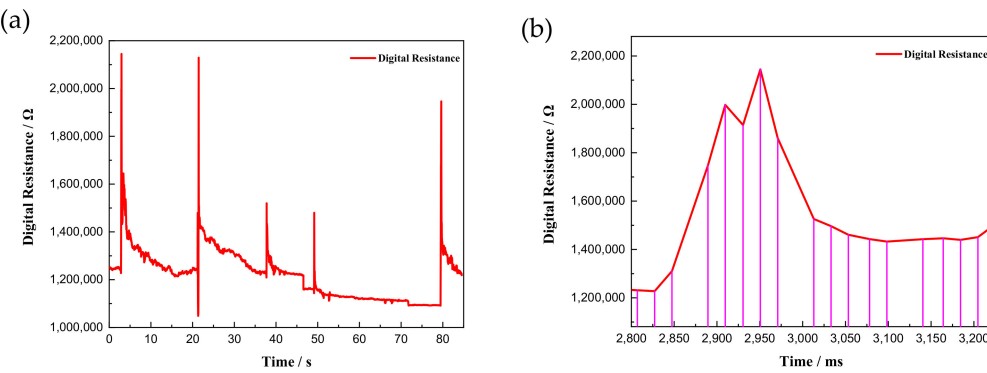

**Figure 15.** (**a**) Line chart between resistance and time; (**b**) recovery time.

Given the reaction limits of the human body, it is unlikely that a boxer could perform two strikes in approximately 100 ms. Therefore, considering the two performance indicators (i.e., response and recovery times), the flexible sensor prepared in this study can fully meet the real-time feedback of the instantaneous strike force during boxing matches.

### 4.2.3. Force Curve

Figure 16 indicates that the digital resistance of the flexible sensor increased with an increase in the punching force. This curve was fitted and a confidence interval analysis was performed using Origin; the linear equation between the digital resistance and force is given in Equation (2):

$$F = 0.9567x + 3351.5 \qquad (2)$$

The R$^2$ of the fitted curve was 0.9421, indicating that the fitted curve agreed well with the experimental curve; almost all data points fall within the 95% confidence interval. Using the fitting equation, the minimum resolution of the portable smart boxing glove was calculated as 1.05 kg.

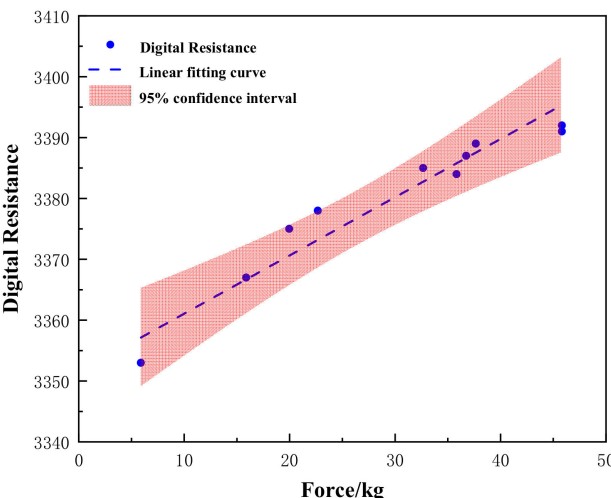

**Figure 16.** Relationship between digital resistance and force.

## 5. Conclusions

A new fabrication process for flexible sensors; i.e., a plasma spraying method, was proposed in this study. The effects of plasma treatment and pattern parameters on the sensitivity, cyclic stability, and strain range were explored. The fabricated sensor was installed on a boxing glove to test it under real conditions and avoid the previous practice of indirectly reflecting the hitting state of the boxer using a rigid acceleration sensor. The following are the conclusions of this study:

(1) A novel plasma spraying method was proposed to fabricate flexible sensors, achieving high reproducibility of the sensor patterns;

(2) The results were compared to those of an untreated sensor. The sensitivity of the flexible strain sensor could be increased from 3.9 to 11.5 when using the plasma treatment (an increase of 195%). However, the strain range was reduced to 10%, making the developed sensor only suitable for small-strain applications;

(3) The effects of the pattern parameters on the sensitivity of the flexible sensor were explored. Multiple repeated paths could also increase the sensitivity of the sensor, but all paths had the same sensitivity;

(4) A smart boxing glove that could transmit and monitor a striking force of 5–50 kg in real time was developed, showing a recovery time of 100 ms and a sensitivity of up to 1.05 kg;

(5) The Wi-Fi data-transmission function of the ESP32 development board was used for real-time data collection and image display, which greatly improved the portability of the smart glove and would not affect the daily and competition use of the glove by boxing enthusiasts and athletes.

In a future study, we will explore additional types of particles available for spraying, as well as the spraying of ultracomplex patterns. In addition, we will combine our sensor-manufacturing methods with more applications. A kinematic model of the exoskeleton may be the basis for a physical model and the analysis of gait dynamics [27]. Our flexible sensors have a promising future in gathering data more accurately and quickly. The sensors manufactured through our plasma spraying method also have a broad application potential in soft robotics and many other fields.

**Author Contributions:** Conceptualization, Y.L., Z.J. and Y.Y.; methodology, Z.J.; software, Y.C. and Z.Z.; validation, Y.L. and Y.C.; formal analysis, Z.J.; investigation, Z.J. and R.L.; resources, Y.L. and Z.J.; data curation, Y.C., R.L. and Z.Z.; writing—original draft preparation, Y.C. and Z.Z.; writing—review and editing, Y.Y., R.W. and Z.J.; visualization, Y.Y., R.W.; supervision, Z.J.; project administration, Y.L. and Z.J.; funding acquisition, Y.L. All authors have read and agreed to the published version of the manuscript.

**Funding:** The work was supported by the Hunan Provincial Department of Education Scientific Research Project Youth Project (No. 21B0612) and Beijing Nova Program from Beijing Municipal Science & Technology Commission (Z201100006820146).

**Conflicts of Interest:** The authors declare no conflict of interest.

## Abbreviations

| | |
|---|---|
| EGaIn | Eutectic gallium–indium |
| TPU | Thermoplastic polyurethane |
| CNT | Carbon nanotube |
| MWCNT | Multiwalled carbon nanotube |
| MMAD | Mass median aerodynamic diameter |
| GF | Gauge factor |
| AP | Access point |
| IP | Internet Protocol |

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
