# Peer review of "Plasma-Sprayed Flexible Strain Sensor and Its Applications in Boxing Glove"

_applsci, doi:10.3390/app12168382_

Round 1

Reviewer 1 Report

The manuscript entitled Plasma-Sprayed Flexible Strain Sensor and Its Applications in Boxing Glove

 submitted by the group of Authors represents a research on the use of novel controllable low-temperature plasma spraying method of conducting nanoparticles to produce flexible strain sensor. The sensor was fabricated on the silica gel base, and the effects of plasma treatment on the cyclic stability and sensitivity of the sensor were examined and compared, pattern parameters were optimized and boxing glove prototype was developed and tested.

The Authors should discuss more about the pattern issues and potential solutions.

The discussion about why the proposed sensing layer fabrication technology was chosen to manufacture patterns? And potential suggestions on other comparable technologies should be discussed.

Does the selection of certain type of CNT (like MWCNT, SWCNT, etc.) make any potential difference in signal stability and properties in general?

Introduction part should be enriched with more recent references.

In general, the manuscript is interesting, the novel spraying method was successfully applied and tested for sensor fabrication, the results were well elaborated and the sensor showed good results in real application.

Author Response

Thanks for your great advice.We have answered the questions one by one according to your requirements, and have carefully revised the article.

Reviewer 2 Report

1.       In the introduction section at line 51, the author has described insufficient bonding force between sensing layer and substrate in the past literature. But the author has chosen 31.5ËšC, which is very low enough to do thermosetting. Has the author gone for substrate study with characterization and morphology at this temperature?

2.       Under results and discussion at line 256, author has mentioned about bond between carbon-carbon atoms, that is broken due to plasma treatment at low temperature. Has the author done any experiment or proof to show this? Any bond energy released due to breakage? How could the author claim this or any literature behind this or reference link?

3.       In figure 10, it has been shown regarding the response of sensors at low-temperature plasma treating that is at 31.5ËšC. How come this temperature is much enough to show good sensitivity and reliability in breaking carbon bonds?

4.       In figure 15, has the author done any error analysis? Digital resistance can be claimed with good error data?

5.       In section 3.3.1, tensile test setup, what is the tensile strain rate or displacement rate been used to do the test?

6.       In the figure3, of the interface for 3D motion control software, what is the step speed or increment rate been used for steeper motion? Please mention it in the article? Also, mention the spray parameters used for fabrication?

Author Response

(The authors gave the same response as above.)

Reviewer 3 Report

Dear Authors,
thank you very much for sending the article titled Plasma-Sprayed Flexible Strain Sensor and Its Applications in Boxing Glove for review process. Looking at the title for the first time, I began reading the article with interest.  The approach of the study appears very original. The contents of the manuscript are quite interesting by his methodology and the tools used. Below are suggestions to the authors:

- please include a Flowchart at the beginning of Materials and Methods section

- please correct typos (line 239) you write big Delta, whereas in Eq.1 is small letter. the same situation for R0. In my opinion authors should read carefully article and correct errors.

- figures should be corrected (I suggest include the vector figures like .eps). It is difficult read axis description (e.g. Fig. 10,11) 

- Fig. 11b what is statistical difference between 1, 5, and 3% (lack of description in the figure (a, b)

- the experiment results should be compared with the literature data 

- lack of future research. Authors should describe what the next research should be done

- very poor state of the art. The idea of use this type of equipment is interesting. For example in the article titled: A kinematic model of a humanoid lower limb exoskeleton with pneumatic actuators, DOI: 10.37190/ABB-01991-2021-05 authors use IMU for obtain kinematic parameters of gait. If they use plasma-sprayed flexible strain sensor the can reduce weight of equipment. To improve article quality I suggest cite above manuscript and write a few sentences in Conclusion. Like I wrote, the idea is interesting, but authors should highlight the advantages of this kind of sensors.

- please include the abbreviation list before References

Author Response

(The authors gave the same response as above.)

Round 2

Reviewer 2 Report

It is ok.

Reviewer 3 Report

Dear Authors,

thank you very much for resending the article titled: Plasma-Sprayed Flexible Strain Sensor and Its Applications in Boxing Glove. In my opinion article in this version can be accepted for publication in Applied Sciences.